# Data on Utility in Cost–Utility Analyses of Genetic Screen-and-Treat Strategies for Breast and Ovarian Cancer

**DOI:** 10.3390/cancers13194879

**Published:** 2021-09-29

**Authors:** Julia Simões Corrêa Galendi, Vera Vennedey, Hannah Kentenich, Stephanie Stock, Dirk Müller

**Affiliations:** Institute of Health Economics and Clinical Epidemiology, Faculty of Medicine and University Hospital of Cologne, University of Cologne, Gleueler Str. 174-176, 50923 Cologne, Germany; vera.vennedey@uk-koeln.de (V.V.); hannah.kentenich@uk-koeln.de (H.K.); stephanie.stock@uk-koeln.de (S.S.); dirk.mueller@uk-koeln.de (D.M.)

**Keywords:** breast cancer, risk assessment, utilities, cost–utility analysis, health preferences, quality of life

## Abstract

**Simple Summary:**

The prevention of hereditary breast and ovarian cancer involves genetic counselling and several highly preference-sensitive alternatives (i.e., risk-reducing surgeries). In health economics models, data on health preferences applied (i.e., utility values) are heterogeneous. In this methodological analysis, we compared the application of utility values among cost–utility models of targeted genetic testing for the prevention of breast and ovarian cancer. While varying utilities on risk-reducing surgeries and cancer states did not impact the cost–utility ratio, utility losses/gains due to a positive/negative test may strongly affect the cost–utility ratio and should be considered mandatory in future models. Because women’s health preferences may have changed as a result of improved oncologic care and genetic counselling, studies for ascertaining women’s health preferences should be updated.

**Abstract:**

Genetic screen-and-treat strategies for the risk-reduction of breast cancer (BC) and ovarian cancer (OC) are often evaluated by cost–utility analyses (CUAs). This analysis compares data on health preferences (i.e., utility values) in CUAs of targeted genetic testing for BC and OC. Based on utilities applied in fourteen CUAs, data on utility including related assumptions were extracted for the health states: (i) genetic test, (ii) risk-reducing surgeries, (iii) BC/OC and (iv) post cancer. In addition, information about the sources of utility and the impact on the cost-effectiveness was extracted. Utility for CUAs relied on heterogeneous data and assumptions for all health states. The utility values ranged from 0.68 to 0.97 for risk-reducing surgeries, 0.6 to 0.85 for BC and 0.5 to 0.82 for OC. In two out of nine studies, considering the impact of the test result strongly affected the cost–effectiveness ratio. While in general utilities seem not to affect the cost–utility ratio, in future modeling studies the impact of a positive/negative test on utility should be considered mandatory. Women’s health preferences, which may have changed as a result of improved oncologic care and genetic counselling, should be re-evaluated.

## 1. Introduction

Several approaches for preventing hereditary breast and ovarian cancer (BC and OC) have been developed in the last decades. Following genetic testing and counselling, women can be offered risk-reducing strategies that significantly reduce the risk of BC/OC and cancer-related mortality. Risk-reducing mastectomy (RRM) has shown to reduce the risk of BC and risk-reducing salpingo-oophorectomy (RRSO) to reduce the risk of both BC and OC [1,2]. However, by opting for each of these risk-reducing surgeries, a woman’s quality of life may be impacted by loss of fertility, premature menopause, suffering due to breast removal and breast reconstruction surgeries [3,4]. 

Because genetic screen-and-treat strategies often proceeded directly from preliminary validation into clinical practice—without a deeper knowledge about the economic implications—credible economic analyses were claimed for [5]. In recent years, health economic models evaluating screen-and-treat strategies for healthy women at increased risk of cancer have been published, most of them conducted as cost–utility analyses (CUA) [6]. However, the contribution of health economic analyses to decision making in health care depends on a country’s health policy and different system level aspects.

A CUA is a specific case of cost-effectiveness analysis, in which the outcomes are generic as opposed to disease-specific and incorporate the notion of value [7]. The concept of value allows for making judgments within a context of constrained resources and is particularly advantageous for comparisons across clinical areas or between heterogeneous interventions (e.g., a drug treatment for heart failure with a lifestyle program for preventing obesity) [8]. Although this approach has attracted ethical and methodological criticism, it has been adopted by the regulatory agencies of many countries (e.g., the UK, Sweden and The Netherlands) [9].

The cost–utility ratio can be obtained after determining an intervention’s additional costs and benefit, the latter valued as utilities. The utilities reflect an individual’s health preferences by transferring all positive and negative effects of an intervention into one numerical value. The quality-adjusted life years (QALYs) incorporates both length and quality of life into a scale of 0 to 1 (i.e., from dead to one year lived with perfect quality of life). Utilities for a CUA can be directly obtained from preference-based elicitation methods such as standard gamble (SG) or time trade-off (TTO), or indirectly by mapping preferences onto the utility scale via a generic health related quality of life questionnaire (e.g., the EuroQol-5-Dimension [EQ-5D]) [10,11,12].

The choice of utility parameters for CUAs of genetic-based screen-and-treat strategies for BC and OC is notably challenging. Different phases of the diseases have to be accounted for, as have sensitive determinants of a woman’s utility such as fear of cancer, burden of cancer, or regret and relief after preventive treatments. Although utility is one among several input parameters (i.e., clinical, epidemiological or cost data) that may affect the cost–utility ratio of CUAs for cancer, utility parameters have often been shown to be especially sensitive, leading to considerable uncertainty [13,14].

The present methodological analysis explores the complex application of utility parameters into health economic models of genetic-based screen-and-treat strategies for BC/OC. The objective was (i) to assess the choice of input data and assumptions for applying QALYs within selected CUAs, and (ii) to assess the degree of uncertainty resulting from utility data.

## 2. Methods

The CUAs for this analysis were obtained from the most recently published systematic review of health economic modeling studies that compared targeted genetic testing for the prevention of BC and OC with no testing [6]. In short, health economic modeling studies that addressed women at increased hereditary risk of BC/OC were included. Targeted women had to be offered genetic testing for inheritable germline mutations, including but not limited to BRCA, with one or more subsequent surgical or nonsurgical treatment options. The search was performed in MEDLINE (via PubMed) and the Centre for Reviews and Dissemination (CRD) on 31 July 2020. 

From this review, modeling studies that provided the outcome as a cost–utility ratio were selected. To ensure a high level of methodological quality, two reviewers independently extracted relevant data on utility application and, in case of disagreement, a consensus meeting was conducted. In cases of missing values, the authors were contacted.

Utility data applied into four health states often used for economic modeling studies of genetic testing for BC and OC were extracted: (i) the test result, (ii) the risk-reducing surgeries (i.e., RRM and RRSO), (iii) BC or OC, and (iv) the post cancer state. In state (i) a woman’s utility can be affected either positively or negatively depending on the test result. Similarly, in state (ii) risk-reducing surgeries may have mixed effects on utility, as a woman’s utility may decrease due to the burden of surgery but may increase as a result of relief from cancer risk. Whereas BC and OC decrease a woman’s utility in the beginning (iii), there is uncertainty how long a woman’s utility is affected in the post cancer state (iv). Similarly, in the post cancer state, a woman’s utility may increase (by reduced physical impairment) or decrease (by ongoing psychological distress due to fear of recurrent cancer). Although cost–utility models usually address both cancer types (i.e., because carriers of BRCA1/2 are at high risk for both BC and OC), most models separate BC and OC as different health states, each with unique utilities and costs associated.

From the CUAs, the source of utility values and the assumptions made were extracted. For sources of utilities referenced by the included studies, it was evaluated if these (i) had elicited direct preference ratings, (ii) had mapped preferences onto the utility scale indirectly via a generic health-related quality of life questionnaire or (iii) were based on other methods (e.g., questionnaires not intended for mapping preferences). In addition, we extracted how utility values were combined in the model. The three methods typically used to combine utilities when data are available only for different single conditions are the additive, the multiplicative or the minimum method [15]. The disclosure of the method for combining utility is important because methodological differences may result in inconsistent policy decision making [16].

In addition, we extracted results from deterministic sensitivity analyses to assess the impact of utility data on the cost–utility ratio. Lastly, the quality of reporting was assessed using the ISPOR SpRUCE checklist, which details in 15 items the desired reporting standards for applying utilities in cost–utility models [17]. 

## 3. Results

The analysis was based on fourteen cost–utility models. The systematic review, upon which the study selection relied, detailed the search strategy, the inclusion criteria, the characteristics of studies concerning the perspective chosen and the input parameters applied [6]. In eight of the fourteen cost–utility models, the quality of reporting was sufficient for the majority of the items, while two items were fulfilled by no study. Table 1 details the assessment for each of the ISPOR SpRUCE checklist items. 

The fourteen cost–utility models included for this analysis differed markedly where the application of utility data is concerned. Table 2 summarizes the application of utility for the selected studies. For all models except four [18,19,20,21], an adjustment for both age and target population was made. For combining data on utility, in six modeling studies the authors declared to have used the multiplicative method [18,19,22,23,24,25], while the remaining studies did not report the method.

### 3.1. Utility Data and Assumptions

Utility values applied to the (i) test, (ii) prophylactic surgery, (iii) cancer states and (iv) post cancer states are illustrated in Figure 1A,B. In the base case, a decrease in utility due to a positive test result was included in five studies (35%) [18,19,22,26,27]. Although Eccleston et al., Moya-Alarcón et al. and Hurry et al. did not apply disutility due to a BRCA mutation in the base case, a potential decrease in utility was assessed in a sensitivity analysis [23,24,25]. The time the disutility from a positive test result persisted ranged from one year [26,28] to over five years [19,29], to a persistent decrease [18]. Holland et al. was the only study that assumed a small utility increase (0.01) for five years due to the relief of receiving a negative test result [29].

Disutility resulting from (ii) prophylactic surgery was considered in all studies except one [31]. Tuffaha et al. assigned no disutility from surgery and justified that the disutility was assumed to be offset by the assurance attained from the reduction in cancer risk [31]. In the other studies, the decrease in utility ranged from 0.03 to 0.14 for risk-reducing mastectomy (RRM) and from 0.003 to 0.24 for risk-reducing salpingo-oophorectomy (RRSO). Eight studies (57%) considered the possibility of performing both surgeries. In eight studies, the duration of the utility decrease was assumed to be short-term (i.e., one year) [23,24,25,26,27], or limited to four or more years [18,19,29].

With regard to (iii) the cancer states, heterogeneity was observed for the consideration or non-consideration of disutility related to a specific stage of cancer (i.e., stages I to IV) or cancer categories for disease progression (e.g., early, advanced or recurrent cancer). The majority of studies reflected disutility by using one state for BC and one state for OC (with two studies adding a combined state for patients with both BC and OC) [21,27]. Two studies provided additional states for metastatic BC and end-stage OC [18,19], while Sun et al. separated utility decreases in early BC/OC, advanced BC/OC, recurrent BC/OC and end-stage OC [20]. The only study that considered different disease stages of BC, was the model of Asphaug et al. [22]. 

The duration a patient’s utility is affected by cancer, often declared as (iv) post cancer states, also varied between the studies. Some studies assumed a permanent disutility from previous BC or OC, while others assumed women to be cured and to regain perfect health. This underlying assumption concerning permanent or transient disutility resulted in final utility values for post cancer ranging from 0.75 to 1 for BC, and from 0.58 to 1 for OC. Due to a lack of long-term data most studies assumed a gradual increase in utility in the post cancer state without achieving the level of utility before cancer [18,19,20,23,24,25,28,29]. A sustained decrease in utility as a result of cancer was assumed in three studies [20,21,30]. In contrast, a constant decrease in utility due to cancer for five years with a complete regain from year six was expected in two studies [26,31].

For all studies with the exception of four [18,19,27,29], the rationales behind the assumptions made for assigning utility to specific health states were unclear.

### 3.2. Sources of Utility Data Reported by Included Studies

All CUAs used data from three or more sources, with the exception of one model [31]. Among all modeling studies, 18 sources were used to model changes in utility due to the test result, risk-reducing surgeries or cancer, with 60% of these studies cited by only one CUA [32,33,34,35,36,37,38,39,40,41,42,43,44,45,46,47,48]. Thirteen studies were published 15 or more years ago [32,33,34,36,37,39,41,43,45,48,49,50,51]. Ten studies were conducted in the U.S. [33,35,37,38,41,45,47,49,52,53], four in the Netherlands [34,48,50,54], three in the U.K., [39,43,46] one in Canada [32], one in Sweden [42], and one in Australia [36]. Sixteen were based on surveys or questionnaires provided to women with BC/OC or at increased risk of cancer, while two studies summarized published evidence in a systematic review [44,47]. Among the references providing primary data, eleven studies conducted direct preference ratings with samples between 31 [34] and 361 [42] participants. Among the direct preference-based approaches, seven applied a TTO approach, [36,37,38,40,42,48,51] three an SG approach [32,43,46] and one both [47]. The sources of utility data are detailed in the Appendix A. 

Two studies elicited utility values for a positive (i) test result, one applying a TTO instrument [38], and the other an SG questionnaire [32]. The TTO-study, which was based on preference ratings of Canadian women who had tested positive for BRCA, was cited by six models [18,19,23,24,27,55]. The SG-based study, which was applied to both high-risk relatives of Canadian women with BC and the general population, was used by Holland et al. for assuming the largest decrease in utilities due to a positive test result (0.17) [32].

Five additional studies were used for justifications of assumptions, four that evaluated the impact of testing positive on a woman’s concerns and psychological status [50,52,53,54], and one that assessed the impact of a negative test result [49]. These were based on surveys/interviews evaluating the impact of genetic testing for BRCA1/2 that used various patient-reported outcome measures (e.g., the Cancer Worry Scale, Hospital Anxiety and Depression Scale) at different time points. Two of these studies were prospective longitudinal surveys evaluating a woman’s psychological status through one year after the test results’ disclosure [52,54], while the third study was a cross-sectional study evaluating the women’s concerns at least four years after disclosure of the results [53]. In contrast, one observational study showed that psychological distress persisted for 5 years [50]. 

Utilities for the state (ii) prophylactic surgery were taken from eight different sources with two sources cited by five studies each. The sources most often referenced were studies presenting TTO preference ratings from the U.S. population for prophylactic surgeries (both RRM and RRSO) [37,38]. Similarly, utilities for health states related to cancer and post cancer states (iii and iv) were each taken from one source preferentially. A systematic review on utilities for BC was cited by ten studies (72%) [44], while a preference rating study using TTO was the source of OC utilities for six studies [40].

### 3.3. Impact of Utility on the Cost-Effectiveness Ratio

Most studies stated the impact of utility data on the cost–utility ratio in one-way sensitivity analyses to be negligible (i.e., as defined by the authors), while three studies did not specifically report how data on utility (or assumptions) had affected the cost-effectiveness ratio [22,24,30]. 

In three studies, the cost-effectiveness ratio was highly affected by utility changes. First, assuming a smaller utility gain from a negative test result (from 0.01 to 0.006) increases the cost–utility ratio strongly or even results in domination of the no-test strategy; in that study, the cost–utility ratio also increased when assuming a larger decrease in utility in the first year after RRM [29]. Second, the consideration of disutility from a positive test result (0.13) increased the cost-effectiveness ratio by 40% [23]. Third, the cost-effectiveness ratio decreased by 50% when assuming no utility decrease resulting from prophylactic surgery at the age of 30 years [21].

## 4. Discussion

The incremental cost and value of offering genetic-based screen-and-treat strategies for women at risk for hereditary BC and OC have often been evaluated by CUAs. When the application of utility parameters is concerned, considerable heterogeneity was observed in the fourteen modeling studies included in this review. The range of utility parameters applied to each health state differed notably between studies. Moreover, our analysis indicates that the choice of utility parameters, particularly those for reflecting a positive genetic test result, may contribute to uncertainty of the model results.

The CUAs relied on different assumptions regarding the effect of genetic testing (i.e., while some assumed a decrease in utility due to a positive test result, others disregarded the possibility of experiencing disutility of testing positive completely or considered it only in a sensitivity analysis). The reason for this discrepancy may be the different sources of utility values, which relied on different study designs and patient-reported outcome measures. According to both non-preference-based surveys and preference-based studies that elicited utility values, a disutility should be accounted for in case of a positive test result [32,38,50,52]. However, these results contrast to two earlier published (non-preference-based) surveys that found no long-term impact of genetic testing for women with confirmed BRCA1/2 mutation [53,54].

The contradictory evidence on how important and how durable the distress of knowingly carrying a genetic mutation to women is, withholds a clear recommendation for modelers on how to reflect a woman’s utility after a positive genetic test. However, psychological distress from a genetic test was a sensitive aspect in two of nine studies considering this aspect [23,29]. Similar to genetic testing, the application of utility values to prophylactic surgeries was also a sensitive aspect in two studies [21,29]. The states (i) and (ii) may affect the cost–utility ratio most strongly because they occur at the beginning of the model where the discount rate is still unreduced, i.e., the exponentiation effect has not yet occurred [56].

A disutility for prophylactic surgery was considered by all studies but one, with values ranging widely from 0.82 to 0.97 for RRM and from 0.68 to 0.97 for RRSO. Utility values for RRSO were taken from two studies presenting TTO preference ratings, one conducted in 1999 and one in 2010 [37,38]. The lowest RRSO utilities were taken from the earlier source (i.e., 0.68–0.78 applied in three of the models [18,29,30]), while the remaining models that applied higher values (e.g., 0.81-0.95) cited mostly the more recent source. The discrepancy between the two TTO preference ratings may reflect that women’s perception of prophylactic surgery have changed with time, e.g., because the surgery technique evolved or because the expertise on genetic counselling has grown, mitigating the disutility of the surgery.

For the genetic testing state and prophylactic surgery, the heterogeneity on sources cited led to relevant discrepancy on the application of utility parameters, while for the (iii) cancer states it did not. In fact, most studies relied on the same preferred sources for the cancer states, and still a wide range of utility values were applied to the models. For cancer states, utility values ranged from 0.6 to 0.85 for BC and from 0.5 to 0.82 for OC. This variability may be justified by the fact that some modelers chose to include advanced stages of the disease (e.g. metastatic and end-stage) within one cancer state, while others opted to create separate health states for advanced stages. Nevertheless, utilities for cancer states had little impact on the cost–utility ratio in the CUAs according to the presented results of sensitivity analyses.

Considering the sources of utility data, various approaches including preference-based ratings, non-preference-based surveys and studies providing data for assumptions on utility were used. Thus, researchers conducting a CUA for a genetic-based screen-and-treat strategy for women at risk for hereditary BC and OC are confronted with this variety. According to recommendations for the identification, review and use of utilities in cost–utility models, many models included in this overview did not provide sufficient information on the suitability and validity of the utilities selected. Although the analysts’ available time and resources may be limited, these should be no reason to take data on utility less seriously than other input parameters [17].

A limitation of our analysis is that the studies were sampled from a previously published systematic review [6], which may have excluded studies published afterwards. However, the considerable degree of heterogeneity in the application of utilities allows to draw some conclusions for future modelling studies. First, alternative approaches to determine how the choice of utility parameters contributed to uncertainty should be considered. In order to determine the impact of a specific parameter on the cost-effectiveness ratio, one-way sensitivity analyses are frequently used. However, Asphaug et al. quantified the uncertainty around utility weights by calculating the value of eliminating parameter uncertainty by partial expected value of perfect information (EVPPI) [22]. The EVPPIs indicate the effect parameters have on the net-benefit difference between the testing strategies, that is, which parameters produce more decision uncertainty (i.e., a high EVPPI is indicative of parameters causing high decision uncertainty) [57]. In the analysis of Asphaug et al., among 23 parameters, utility/QALY-weights was the ninth highest EVPPI value (about USD 100 per woman).

Second, because available preference-based studies provide robust evidence for women’s small but significant disutility from a positive test result, future modeling studies for genetic screen-and-treat strategies to prevent BC/OC should mandatorily apply these utility losses, at least in sensitive analyses. Moreover, the contradictory results of non-preference-based studies, with more recently published studies indicating no distress from a positive genetic test [50,52], should also stimulate further research in this field. These studies may indicate a change in women’s attitudes resulting from improved psycho-oncologic care and genetic counseling. However, these studies were based on only 8% of women being BRCA positive and a lack of variation in genetic test concerns due to a homogeneous sociodemographic study sample [52,53], requiring more and, particularly, preference-based studies.

Third, efforts are needed to validate model structures that aggregate different stages of cancer in one health state. While long-term disutility resulting from cancer does not appear to have a large impact on the cost-effectiveness ratio, the lack of reflecting specific disease stages in models may have an impact on the results. Moreover, utilities obtained from the same source should be applied consistently and the methods of combining utilities provided. In addition, modelers would benefit from efforts to transparently synthesize data from various sources, e.g., systematic reviews such as that from Peasgood et al. [44].

Fourth, efforts should be made to develop a reference model because heterogeneity in modelling studies for assessing preventive strategies for BC/OC was not limited to utilities but also observed for other aspects (e.g., model structure, tools used for collecting clinical and epidemiological data, methods of validation) [6]. As demonstrated for other diseases, a reference model ensures new models will be populated based on the systematic identification of the best available data [58,59]. This would increase the likelihood of models to be consistent and comparable across different countries.

Finally, a revision of the current knowledge on health preferences should be encouraged. The utilities applied in health economic models probably reflect outdated disease experiences that may have changed because of improved oncologic care, the provision of genetic counselling and women’s attitudes to preventive surgery. Therefore, regularly updated surveys for adjusting utilities (e.g., within clinical trials) are needed.

## 5. Conclusions

While, in general, utilities applied to breast and ovarian cancer health states have not been shown to be a sensitive parameter for the cost–utility ratio, in future modeling studies the impact of a positive/negative test on utility should be considered mandatory. In addition, analysts should be aware that the literature on women’s health preferences might be outdated due to improvements in oncologic care in the last decades. Hence, surveys for re-estimating utilities would be useful for identifying potential changes in women’s health preferences. 

## Figures and Tables

**Figure 1 cancers-13-04879-f001:**
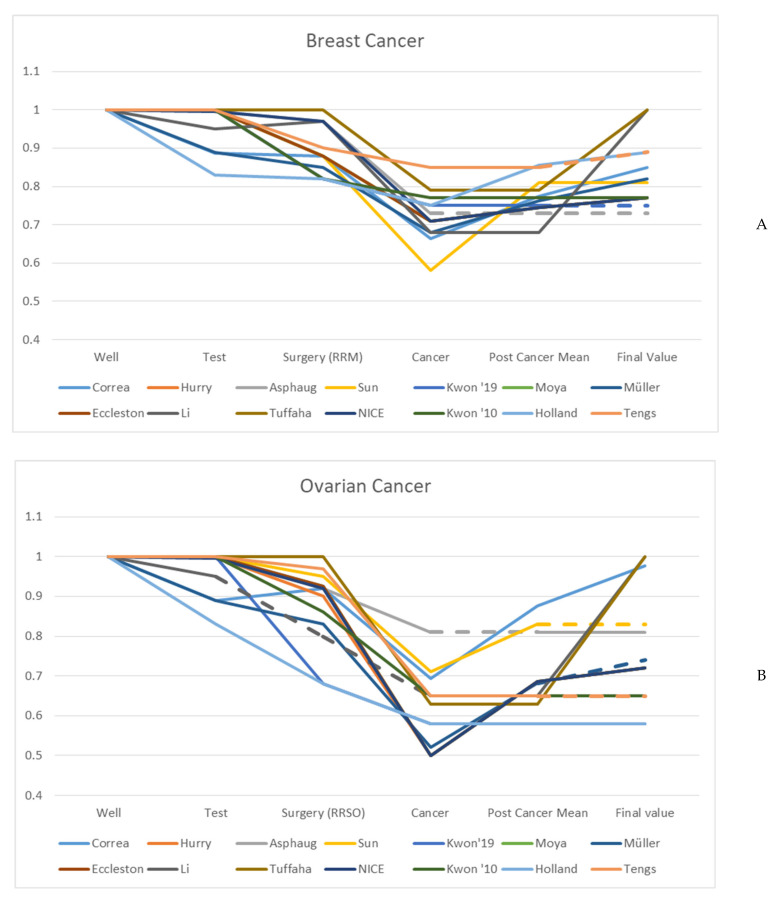
Changes in utility values applied to the cost–utility models ((i.) genetic testing; (ii.) risk-reducing surgery; (iii.) cancer states and (iv.) post cancer states). In (**A**) risk-reducing mastectomy (RRM) and breast cancer and (**B**) risk-reducing salpingo-oophorectomy (RRSO) and ovarian cancer. Dashed lines represent missing values (i.e., Asphaug et al. did not report values applied to post cancer states [22]; Tengs et al. and Kwon et al. (Kwon’19) did not consider a post cancer state in the model [21,30]; finally, Li et al. did not consider oophorectomy as a risk-reducing surgery in the model) [26].

**Table 1 cancers-13-04879-t001:** Quality of reporting of utilities by the included cost–utility models, assessed using the ISPOR SpRUCE checklist.

	Search Terms and Scope	Study Selection Criteria	Quality Check	Assessment of HSU Relevance	Population Characteristics	Measure Used	Preference Weights	Descriptive Statistics about HSUs	Response Rate for the Measure Used	Extent of Missing Data or Lost to Follow-Up	Original Reference	Basis for Selecting HSUs	Method Used to Combine Estimates	Actual HSUs Used	Adjustments/Assumptions
Simoes Correa Galendi et al. (Brazil, 2020)	✅	✅	✅	✅	✅	✅	✅	✅	❌	❌	✅	✅	✅	✅	✅
Hurry et al. (Canada, 2020)	❌	✅	✅	✅	❌	❌	✅	✅	❌	❌	✅	✅	✅	✅	✅
Asphaug et al. (Norway, 2019)	❌	❌	❌	✅	❌	✅	❌	✅	❌	❌	⚠	✅	✅	✅	⚠
Sun et al. (U.S., 2019)	❌	❌	❌	❌	❌	❌	❌	⚠	❌	❌	✅	⚠	❌	✅	❌
Kwon et al. (Canada, 2019)	❌	❌	❌	✅	❌	❌	❌	✅	❌	❌	⚠	❌	❌	✅	✅
Moya-Alarcón et al. (Spain, 2019)	⚠	✅	✅	✅	⚠	❌	❌	✅	❌	❌	✅	✅	✅	✅	✅
Müller et al. (Germany, 2019)	✅	✅	✅	✅	✅	✅	✅	✅	❌	❌	✅	✅	✅	✅	✅
Eccleston et al. (U.K., 2017)	NA	NA	✅	✅	✅	❌	❌	✅	❌	❌	✅	✅	✅	✅	✅
Li et al. (U.S., 2017)	❌	❌	❌	❌	❌	❌	❌	⚠	❌	❌	❌	❌	❌	✅	⚠
Tuffaha et al. (Australia, 2017)	NA	NA	✅	✅	✅	✅	✅	✅	❌	❌	✅	✅	❌	✅	✅
NICE (U.K., 2013)	✅	✅	✅	✅	✅	✅	✅	✅	❌	❌	✅	✅	✅	✅	✅
Kwon et al. (Canada, 2010)	❌	❌	❌	❌	❌	❌	❌	✅	❌	❌	⚠	✅	❌	✅	✅
Holland et al. (U.S., 2009)	❌	✅	✅	✅	✅	✅	✅	✅	❌	❌	✅	✅	❌	✅	✅
Tengs et al. (U.S., 2000)	❌	❌	❌	⚠	❌	❌	❌	⚠	❌	❌	✅	⚠	✅	✅	⚠

Legend: ✅: reported; ❌: not reported; ⚠: partially reported; NA: not applicable (i.e., primary data or primarily recommended source with no need for literature search).

**Table 2 cancers-13-04879-t002:** Overview of the utility values applied into the cost–utility models and the impact of utility input data on one-way sensitivity analysis.

Study	Utilities and Assumptions				Adjustment	Impact in One-Way Sensitivity Analysis
	(i) Test Positive	(ii) Prophylactic Surgery	(iii) Cancer	(iv) Post Cancer		
Simoes Correa Galendi et al. (Brazil, 2020)	Complete regain within 4 years: 0.89	Complete regain ^b^ within 4 years: RRM: 0.88 RRSO: 0.92	BC: 0.66 Metastatic BC: 0.64 OC: 0.69 end-stage OC: 0.55	Partial regain within 5 years linearly: BC: 0.77 OC: 0.72	Age	Negligible ^c^
Hurry et al. (Canada, 2020)	1.00 ^a^	Disutility for one year: RRM: 0.88 RRSO: 0.95 Both interventions: 0.84	BC: 0.71 OC: 0.50	Partial regain within 5 years: BC: 0.77 OC: 0.72	Age and target population	Not reported
Asphaug et al. (Norway, 2019)	0.995	RRM: 0.97 RRSO: 0.92 Both interventions: 0.89	BC stage I/II: 0.73 BC stage III/IV: 0.55 OC, local: 0.81 Metastatic OC (regional): 0.55 Metastatic OC (distal): 0.16	Not reported	Age and target population	Not reported
Sun et al. (United States, 2019)	-	RRM: 0.88 RRSO: 0.95	Early BC/OC: 0.71/0.81 Advanced BC/OC: 0.65/0.55 Recurrent BC/OC: 0.45/0.50 End-stage OC: 0.16	Partial and sustained regain: BC: 0.81 OC: 0.83	None	Negligible
Kwon et al. (Canada, 2019)	-	RRM: 0.82 RRSO: 0.68	BC: 0.75 OC: 0.58	Sustained decrease, as in (iii)	Age and target population	Not reported
Moya-Alarcón et al. (Spain, 2019)	1.00 ^a^	Disutility for one year: RRM: 0.88 RRSO: 0.95 Both interventions: 0.84	BC: 0.71 OC: 0.50	Partial regain within 5 years linearly: BC: 0.77 OC: 0.72	Age and target population	ICER changed by +/− 10% when varying cancer utilities
Müller et al. (Germany, 2019)	Persistent decrease: 0.89	Complete regain ^b^ within 4 years: RRM: 0.85 RRSO: 0.83 Both interventions: 0.78	Early BC: 0.68 Metastatic BC: 0.63 Early OC: 0.52 End-stage OC: 0.16	Partial regain within 5 years linearly: BC: 0.79 OC: 0.74 Sustained decrease from metastatic BC	Age	Negligible
Eccleston et al. (United Kingdom, 2017)	1.00 ^a^	Disutility for one year: RRM: 0.88 RRSO: 0.95 Both interventions: 0.16	BC: 0.71 OC: 0.50	Partial regain within 5 years linearly: BC: 0.77 OC: 0.72	Age and target population	Consideration of disutility due to a positive test result (−0.13) increased ICER by 40%
Li et al. (United States, 2017)	Disutility for one year: 0.95	Disutility for one year: RRM: 0.97	BC: 0.68 OC: 0.65	Persistent decrease for 5 years and complete regain after on: BC: 0.68 OC: 0.65	Age and target population	Negligible
Tuffaha et al. (Australia, 2017)	-	-	BC: 0.79 OC: 0.63	Persistent decrease for 5 years and complete regain after on: BC: 0.79 OC: 0.63	Age and target population	Negligible
NICE (United Kingdom, 2013)	Disutility for one year: 0.995	Disutility for one year: RRM: 0.97 RRSO: 0.92 Both interventions: 0.89	BC: 0.71 OC: 0.50	Partial regain within 5 years linearly: BC: 0.77 OC: 0.72	Age and target population	Negligible
Kwon et al. (Canada, 2010)	-	RRM: 0.82 RRSO: 0.86 Both interventions: 0.79	BC: 0.77 OC: 0.65	Sustained decrease, as in (iii)	Age and target population	Negligible
Holland et al. (United States, 2009)	Complete regain within 5 years: 0.83	Complete regain ^b^ at age 60: RRM: 0.82 RRSO: 0.68	BC: 0.75 OC: 0.71	Partial regain for BC within 3 years linearly: BC: 0.89 OC: 0.58 (sustained)	Age and target population	Sensitive to test result and prophylactic surgery
Tengs et al. (United States, 2000)	-	RRM: 0.86 RRSO: 0.81 ^b^ Both interventions: 0.86	BC: 0.89 OC: 0.82 Both: 0.82	Sustained decrease, as in (iii)	None	ICER decreases by 50% when assuming no impact from prophylactic surgery

^a^ Lower value tested in sensitivity analysis; ^b^ only assumed for women <50 years with hormone replacement therapy (otherwise: no change). Abbreviations: RRM: Risk-reducing mastectomy; RRSO: Risk-reducing salpingo-oophorectomy; BC: Breast cancer; OC: Ovarian cancer; ICER: Incremental cost-effectiveness ratio. ^b^ to previous health state. ^c^ as defined by authors.

## Data Availability

The data presented in this study are available in its manuscript and Appendix A.

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
