# Peer review of "Data on Utility in Cost–Utility Analyses of Genetic Screen-and-Treat Strategies for Breast and Ovarian Cancer"

_cancers, 2021, doi:10.3390/cancers13194879_

Round 1

Reviewer 1 Report

This article is meta-analysis of cost-utility analysis of the risk-reduction of breast cancer (BC) and ovarian cancer (OC). This article can be a good guidance to know the cost-utility analysis of BC and OC.

Author Response

Thank you for the overall positive evaluation of the manuscript.

Reviewer 2 Report

First, the manuscript is well written but I am not sure if the content meets the scope of the Journal (‘high-quality articles including basic, translational, and clinical studies on all tumor types’). I think the manuscript is too superficial with no in depth analysis. It seems a general review with no information on the data evaluation and statistics. At the same time, table and figure captions are very scant which makes difficult to follow the idea of the analysis. I am not sure if the two presented cancer types should be considered together since varying  women’s health preferences appear. I would rather suggest separating BC and OC when assessing cost-utility analyses.

Considering the text body – the summary and intro seem not very conclusive.

Author Response

Thank you for thoroughly reviewing the manuscript. In this article we reviewed how women’s’ health preferences have been depicted in cost-utility models assessing screen-and-treat strategies to the prevention of hereditary breast and ovarian cancer. By publishing this paper in a journal with a clinical focus we hope to call the attention of practitioners and to stimulate them to consider updating current data on health preferences (i.e., utilities), which could be used for future health economic models.

The figure and table captions have been updated to better describe their content. Table 1 present an overview of the extracted data, providing more clarity on which utility data was applied to each model. Figure 1 shows how the utility values change as a women move from one health state to another within the models. 

Hereditary breast and ovarian cancer (HBOC) are caused by the pathogenic variant on the BRCA 1 or BRCA 2 genes. The interventions proposed to HBOC consist of genetic testing for both variants, followed by risk reducing strategies. Hence, cost-utility models usually address both cancer types because carriers of BRCA1/2 are at high risk for both breast and ovarian cancer. Moreover, risk-reducing salpingo-oophorectomy reduces both breast and ovarian cancer risk. These two cancer types are depicted as separate health states in the models, with unique utilities and costs associated. With the intent to clarify this point, we included this rationale on lines 39-41 of the introduction and lines 98-100 of the Methods section.

Reviewer 3 Report

1.The present study used complex application of utility parameters into health economic models of genetic-based screen-and-treat strategies for  BC/OC. The methodological analysis is appropriate and can provide useful information for the policy analyses of the disease screening and detection.

2. It is difficulty to evaluate the completeness of the data collection and make sure the quality of each data collection, the author should have a objective briefing to these data.

3. The screening and treatment aspects are related to the health policy and health insurance system of each country, and how to make sure these factors are considered in the current analyses.

Author Response

Thank you for the overall positive evaluation of the manuscript.  

With regard to the completeness of data collection, we agree that an objective briefing would enhance the analysis. To address this, we applied the ISPOR SpRUCE checklist to assess the reporting standards of utility values on the included cost-utility models (paragraph included on lines 111-118 and table 1). However, even the authors of the checklist recognize that these are desired standards that frequently are not feasible due to word restrictions in most journals. For the sake of this analysis, we complemented the data collection looking at the sources of utility data, to extract data on the population’s characteristics and methods.

To comment 3: we agree with the reviewer that policy and system aspects are pre-conditions for the acceptance of cost-effectiveness analyses, as some countries (e.g. UK) rely on them, and others not (e.g., Germany). This aspect was included in the introduction (introduction, page 2, line 49/51). Besides, when relying on cost-utility analyses, a central aspect is to obtain utility data from patients living in the target country.

Round 2

Reviewer 2 Report

I accept the present form.

Author Response

Thank you for reviewing the manuscript.